# Embryonic Thermal Manipulation Affects Body Performance Parameters and Cecum Microbiome in Broiler Chickens in Response to Post-Hatch Chronic Heat Stress Challenge

**DOI:** 10.3390/ani15121677

**Published:** 2025-06-06

**Authors:** Rahmeh Dahadha, Seif Hundam, Mohammad Borhan Al-Zghoul, Lo’ai Alanagreh, Mustafa Ababneh, Mohammad Mayyas, Daoud Alghizzawi, Minas A. Mustafa, David E. Gerrard, Rami A. Dalloul

**Affiliations:** 1Department of Basic Medical Veterinary Sciences, Faculty of Veterinary Medicine, Jordan University of Science and Technology, Irbid 22110, Jordan; rmdahadha23@vet.just.edu.jo (R.D.); sfhundam23@vet.just.edu.jo (S.H.); ababnem@just.edu.jo (M.A.); daalghizzawi20@vet.just.edu.jo (D.A.); 2Department of Medical Laboratory Sciences, Faculty of Applied Medical Sciences, The Hashemite University, Zarqa 13133, Jordan; loai-alanagreh@hu.edu.jo (L.A.); minasahmad36@gmail.com (M.A.M.); 3Department of Medical Laboratory Sciences, Faculty of Allied Medical Sciences, Zarqa University, Zarqa 13110, Jordan; 4Department of Animal Production, Faculty of Agriculture, Jordan University of Science and Technology, Irbid 22110, Jordan; mohammad.mayyas@jsmo.gov.jo; 5School of Animal Sciences, Virginia Polytechnic Institute and State University, Blacksburg, VA 24061, USA; dgerrard@vt.edu; 6Department of Poultry Science, University of Georgia, Athens, GA 30602, USA; rami.dalloul@uga.edu

**Keywords:** broiler, thermal manipulation, microbiome, heat stress, gut health, thermotolerance

## Abstract

Heat stress disrupts the gut microbiota of broilers, which plays a vital role in digestion and immunity. This study investigated whether the thermal manipulation of eggs during incubation could help chickens cope better with heat stress after hatching. Fertile eggs were divided into two groups: one incubated under standard conditions and another exposed to an increased temperature and humidity during specific incubation periods. After hatching, some chicks from each group were placed in heat stress conditions, while others stayed in normal temperatures. We assessed their body weight, body temperature, and gut microbiome composition. Broilers that underwent thermal manipulation displayed a better body weight under heat stress and exhibited alterations in their gut microbiome. These findings suggest that thermal manipulation during incubation could be a simple, cost-effective way to help chickens to acquire thermotolerance, improving their health and growth.

## 1. Introduction

Global warming and the ongoing rise in ambient temperatures pose significant risks to the poultry industry’s future [1]. Heat stress (HS) is when animals’ body temperature rises because they cannot release extra heat into the environment [2]. There are two types of HS in chicken production: acute, which involves a short, abrupt increase in temperature, and chronic, which involves a prolonged rise in the ambient temperature [3].

Among the various physiological systems affected by HS, the gut (gastrointestinal tract) is especially vulnerable [4]. Both acute and chronic ambient temperature fluctuations can significantly impact gut health and function [5]. HS disrupts normal blood flow by promoting peripheral vasodilation and inducing vasoconstriction in the gut, which leads to hypoperfusion. This impaired circulation damages the intestinal mucosa, increasing permeability and allowing bacteria and toxins to enter the bloodstream. Additionally, HS disrupts the gut microbiome, causing dysbiosis. This imbalance compromises the overall health and performance and heightens the risk of pathogenic colonization [5,6,7,8,9].

The gut microbiota refers to the diverse variety of microorganisms that inhabit the digestive tract, including bacteria, fungi, viruses, and protozoa [10]. Among the various regions of the gastrointestinal tract, the cecum harbors the most dense and diverse microbial populations, which play critical roles in immune system regulation, nutrient absorption, digestion, and overall gut health [10]. A balanced gut microbiome protects against pathogenic bacteria and maintains the optimal broiler performance [11]. Disruptions to this microbial balance, such as those caused by HS, can compromise gut integrity and immune function, ultimately reducing growth rates and feed efficiency [12,13].

In chickens, HS significantly alters the gut microbiota, particularly in the cecum. It influences the proportion of key bacterial taxa and potentially disrupts metabolic and intestinal health [14,15]. HS induces a persistent inflammatory status in the gut, weakening the barrier and increasing permeability, negatively affecting the gut microbiome [16,17].

Thermal manipulation (TM), a process of temperature and humidity modification during the critical embryogenesis period, is a highly effective and cost-efficient method for enhancing broiler health and productivity [18,19,20,21]. Multiple research studies indicate that it enhances post-hatch performance by inducing epigenetic changes, achieved through precise adjustments to the incubation temperature and humidity during the critical phase of embryonic development [22,23]. For instance, TM has been associated with improved immune responses under heat stress, improved gut health and immunity in chickens, an increased resistance to bacterial challenges, and enhanced muscle growth [18,19,24,25,26]. Recent studies indicate that significant microbial changes occur during embryonic development, yet the role of embryonic environmental factors, including TM, in shaping microbiomes remains unclear. While HS effects on broiler performance and gut microbiota have been studied, the impact of embryonic TM on post-hatch microbial dynamics is still largely unexplored [13,27].

This study aimed to investigate the effect of TM (39 °C for 18 h per day from embryonic days 10 to 18) on the body weight, body temperature, and cecal microbiome of broiler chickens during subsequent chronic heat stress from post-hatch days 18 to 22. The global shift towards sustainable animal production practices and improving the resilience to environmental stressors, such as heat stress, is essential for promoting healthier livestock. TM has been suggested as a potential strategy to enhance stress resilience and improve gut health. By investigating the interactions between TM, gut microbiota, and heat stress resilience, this research contributes to the development of more sustainable poultry production systems.

## 2. Materials and Methods

The Animal Care and Use Committee of Jordan University and Technology (JUST-ACUC) has accepted all processes performed in the current study (Approval Number: 16/4/12/348). The study used a 2 × 2 factorial design: TM versus control (CON) incubation conditions and chronic heat stress (CHS) versus thermoneutral post-hatch treatments.

### 2.1. Study Population and Incubation

Figure 1 provides a summary of the experimental design of the current study. A total of 800 viable eggs were obtained from 18-week-old Indian River broiler breeder flocks located in Ar-Ramtha, Jordan, sourced from certified and approved suppliers. The acquired eggs were examined for any indicators of damage. Selected eggs (53.8 ± 3.49 g) were incubated into two incubation groups: the TM and the CON groups. Throughout the embryogenesis period, the eggs in the control group were maintained at a consistent temperature of 37.8 °C and 56% relative humidity (RH). The TM group’s eggs were incubated under standard conditions, except that from embryonic days 10 to 18, they were incubated for 18 h per day, from 12:00 PM to 8:00 AM the following morning. Incubation was performed using commercial Masalles Mod.25-I PDS incubators (Masalles, Barcelona, Spain). On the seventh day of incubation, the eggs were examined by candling, and those that contained dead embryos or were infertile were discarded.

### 2.2. Hatching Management and Post-Hatching Rearing

On the day of hatching, chicks were monitored hourly. After hatching, the dried chicks were kept in the animal house at Jordan University of Science and Technology, which served as the site for the research studies. Each chick was feather sexed. Two groups of male broilers, each with an equal number, were formed. Each group contained sixteen cage enclosures (80 cm × 40 cm × 100 cm), with eight chicks per pen. The pen was regarded as an experimental unit (8 chick/pen). Each group was weighed up in the morning. Chicks were fed basal diets as recommended by the National Research Council to fulfill standard nutrient requirements (Table 1), adapted from Shakouri and Malekzadeh [28].

The chicks were raised in two feeding phases: a starter feed from day 1 to day 14 and a grower feed from day 15 to day 22. Their diets provided metabolizable energy levels of 2950 and 2960 kcal/kg and crude protein levels of 21.20% and 18.49%, respectively, offered ad libitum. Within the first five days, the chicks were vaccinated against infectious bronchitis and Newcastle disease. Throughout the field testing, they had unrestricted access to both water and feed. Consistent management practices were applied to all chicks during their rearing period.

The experimental trial lasted for 22 days, from April 10 to May 2. The room temperature was maintained at 33 ± 1 °C for the first week, and by the end of the third week, it had dropped to 24 °C.

### 2.3. CHS Challenge

The CHS challenge was performed on the 18th post-hatch day. Sixty male chicks were randomly selected from both groups (TM and CON) and divided into subgroups for the CHS experiment (*n* = 30 each, with 4 pens per subgroup containing 7–8 birds). The TM and control CHS subgroups (CHS-CON and CHS-TM) were exposed to a constant 35 ± 0.5° C for 5 consecutive days (days 18–22), while non-CHS groups remained at 24 ± 0.5 °C. While broiler heat sensitivity typically peaks around day 28, we initiated CHS at day 18 to evaluate TM-induced adaptations prior to peak sensitivity while avoiding the severe welfare challenges associated with prolonged heat stress in older birds. All environmental conditions and sampling procedures were rigorously standardized across treatment groups.

Body weight (BW) and body temperature (BT) were measured from 20 randomly selected chicks per subgroup on days 18, 20, and 22 post-hatch. A J/K/T thermocouple meter connected to a rat rectal probe (Kent Scientific Corp., Torrington, CT, USA; ±0.1 °C) was used to measure the chickens’ body temperature.

### 2.4. Microbiological Analysis

#### 2.4.1. DNA Isolation and Sequencing

Cecal content samples were gathered for the microbiota investigation on the fifth day of CHS. The chickens were humanely euthanized, and the ceca were aseptically removed using sterile tools. The ceca were carefully and quickly snap-frozen in situ to avoid DNA degradation. To maintain the integrity of the microbial DNA until extraction, the samples were kept at −80 °C in a CryoCube^®^ F570 Series Ultra-Low Temperature (ULT) Freezer (Eppendorf, Hamburg, Germany).

Bacterial DNA was extracted from approximately 250 mg of cecal content using the DNeasy PowerSoil Pro Kit (QIAGEN, Hilden, Germany, CA, USA), following the protocol previously described by Christoff et al. [29]. The (V3-V4) region of the 16S rRNA gene was amplified using universal primers Bakt_341F: (CCTACGGGNGGCWGCAG) and Bakt_805R: (GACTACHVGGGTATCTAATC C). The sequencing was performed commercially by Macrogen Inc., Seoul, Korea, utilizing an Illumina MiSeq platform (Illumina, San Diego, CA, USA) following the 2 × 300 bp paired-end sequencing protocol.

#### 2.4.2. Data Analysis by Bioinformatic Tools

QIIME 2 2024.8 was utilized [30]. The QIIME 2-DADA2 plugin, which offers raw sequence data quality filtering and denoising, was used to apply an amplicon sequence variant (ASV)-based analysis to the 16S rRNA raw sequence data [31]. Following the alignment of all amplicon sequence variants (ASVs) using MAFFT Multiple Sequence Alignment Software Version 7 (via q2-alignment) [32], a phylogeny was built using FastTree 2 (via q2-phylogeny) [33]. ASVs were taxonomized using the q2 feature classifier, which employs the Naive Bayes Silva 138 99% OTUs full-length sequences taxonomy classifier [34,35]. Following the rarefaction (subsampling without replacement) of samples to 23,348 sequences per sample, which were chosen based on the minimum sequencing depth for all samples, q2-diversity was used to calculate rarefaction curves and alpha diversity (within-sample diversity) metrics (Chao1 and Shannon). Using data exported from QIIME 2, bacterial relative abundances were displayed using R (version 4.3.2)and RStudio(version 2024.06.0) [36]. The ampvis2 R package was used for beta diversity ordination analysis [37]. Canonical Correspondence Analysis (CCA) was conducted via the amp_ordinate function. Before CCA, the data underwent Hellinger transformation to standardize the data for ordination [38]. MicrobiomeAnalyst performed PERMANOVA analysis based on the Bray–Curtis dissimilarity measures [39]. The complete bioinformatics workflow for microbiome analysis is presented in Appendix A.

For comparing relative abundances of bacterial taxa, the Kruskal–Wallis H-test was performed using STAMP (version 2.1.3) software, and standard deviations were visualized with error bars [40].

### 2.5. Statistical Analysis

IBM SPSS Statistics 27.0 was used for all statistical analyses (IBM software, Chicago, IL, USA). The study utilized a 2 × 2 factorial design with random sampling at each measurement time point. Treatment group comparisons for BT and BW within the same day were analyzed using two-way ANOVA. Temporal changes in BT within each group (e.g., CON group: day 1 vs. day 2 vs. day 3) were assessed using one-way ANOVA. Data visualization was performed using GraphPad Prism version 10.0 (GraphPad Software, San Diego, CA, USA) for BT and BW. Alpha diversity indexes were compared using a two-way ANOVA and presented as means ± SD. Statistical significance was set at *p* < 0.05 for all analyses.

## 3. Results

### 3.1. TM and CHS Challenge Effects on Body Weight (BW) and Body Temperature (BT)

Figure 2a illustrates changes in BW over time following the CHS challenge. On day 0 of CHS (18 days of age), all groups had comparable initial weights (~540 g). By day 5 post-CHS, statistically significant effects of the TM (*p* = 0.027), CHS (*p* = 0.001), and their interaction (*p* = 0.043) were observed. These findings indicate that both TM and CHS independently influenced the final BW and that their interaction also had a significant impact. At this time point, the non-challenged groups (CON and TM) reached higher final BW values (988.8 g and 978.1 g, respectively), while the CHS-TM group exhibited a significantly greater BW (941.8 g) than the CHS-CON group (842.3 g), suggesting a mitigating effect of TM on the CHS-induced growth reduction.

Figure 2b demonstrates body temperature changes among the experimental groups in response to the CHS challenge. On day 1 post-CHS (19 days of age), a statistically significant effect of the TM (*p* < 0.001), CHS (*p* < 0.001), and their interaction (*p* < 0.001) was observed, with the TM group showing markedly lower BT values compared to the CON group. During days 3 and 5 post-CHS, both CHS-exposed groups (CHS-CON and CHS-TM) exhibited significant BT elevations, which are consistent with the stress response. On day 3, CHS exerted a significant main effect (*p* < 0.001), but TM (*p* = 0.445) and the interaction (*p* = 0.227) were not statistically significant. By day 5, a significant interaction between TM and CHS was detected (*p* = 0.008), although the main effects of TM (*p* = 0.196) and CHS (*p* = 0.367) were not individually significant. Furthermore, the CHS-TM group maintained non-significantly lower BT values than the CHS-CON group on days 3 and 5.

### 3.2. Effects of TM and CHS on Alpha Diversity

Chao1 and Shannon indices were calculated for four experimental groups: CON, TM, and their subgroups challenged with CHS (CHS-CON and CHS-TM) (Table 2). No significant differences were observed between groups for either index (all *p* > 0.05). This suggests that the alpha diversity was not directly impacted by either the TM or CHS challenge in the experimental settings studied. The rarefaction curve analysis, presented in Appendix A, confirmed an adequate sampling depth across all groups.

### 3.3. Effects of TM and CHS on Beta Diversity

Figure 3 illustrates the results of the Canonical Correspondence Analysis (CCA), which visualizes the variation in the microbial community among experimental groups.

The beta diversity was analyzed using the Bray–Curtis dissimilarity index to construct a distance matrix. A pairwise PERMANOVA analysis was then employed to identify significant differences between the groups, with the detailed statistical outcomes in Table 3, with no significant dispersion effects (PERMDISP: F = 2.06, *p* = 0.14), validating the PERMANOVA assumptions.

Pairwise PERMANOVA comparisons indicated no significant differences in the microbial community structure between the CHS-CON and CON (*p* = 0.398) or between the CHS-TM and TM (*p* = 0.08), suggesting a relatively limited influence of CHS on beta diversity. In contrast, significant differences were observed between the TM and CON (*p* = 0.041) and between the TM and CHS-CON (*p* = 0.012), highlighting the more pronounced effect of TM on the microbial community structure.

### 3.4. Effects of TM and CHS on Cecal Microbiota: Phylum-Level Composition

At the phylum level, Firmicutes (66.8–80.7%) and Bacteroidota (18.6–32.7%) were the dominant bacterial phyla across all groups. Proteobacteria, Actinobacteria, and Cyanobacteria made minor contributions (Figure 4a).

Notably, the CHS-TM group exhibited the lowest relative abundance (66.8%) of Firmicutes, while higher levels were observed in the other groups: CON (80%), TM (80.7%), and CHS-CON (78.4%). In contrast, Bacteroidota was higher in the CHS-TM (32.7%) compared with other groups: CON (19.5%), TM (18.6%), and CHS-CON (21.2%). Although these trends suggested a shift in the phylum-level composition, particularly under the combined effects of TM and CHS, none of the differences were statistically significant. The Firmicutes-to-Bacteroidota (F/B) ratio was also calculated for each sample and analyzed using the Kruskal–Wallis H-test, revealing no significant differences across groups (*p* = 0.12).

### 3.5. Effects of TM and CHS on Cecal Microbiota: Class-Level Composition

Figure 4b illustrates the class-level composition of the cecal microbiota across all experimental groups. The most prevalent bacterial classes were Clostridia (49.5–71.4%), Bacteroidia (18.6–32.7%), and Bacilli (9.3–21.7%). Other classes, such as Gammaproteobacteria, Coriobacteriia, and Vampirivibrionia, were present at relatively lower abundances.

CHS influenced the abundance of Bacilli (Figure 5a), with statistically significant differences between the TM- (9.3%) and CHS-challenged groups (CHS-CON: 21.7%; CHS-TM: 17.3%). Although the CON group also exhibited a lower relative abundance of Bacilli (14.4%) compared to the CHS-challenged groups, this difference was not statistically significant.

### 3.6. Effects of TM and CHS on Cecal Microbiota: Order-Level Composition

Figure 4c illustrates the top 10 bacterial orders of the cecal microbiota across all experimental groups. The most prevalent orders were Lachnospirales (18.4–35.6%), Oscillospirales (22.7–39.7%), Bacteroidales (18.6–32.7%), and Lactobacillales (7.7–13.6%). Other orders, such as Erysipelotrichales, Clostridia_UCG_014, Bacillales, Clostridia_vadinBB60_group, Monoglobales, and Enterobacterales, were present at relatively lower abundances.

Lachnospirales displayed a non-significant increase in relative abundance in the CON group (35.6%) compared to the TM group (28.4%). Notably, its abundance was lower in CHS exposure groups (CHS-CON: 30.7%; CHS-TM: 18.4%). However, statistically significant differences were observed in the CHS-TM group compared to the other groups (Figure 5b). These results suggest that TM may uniquely affect the cecal microbiota in response to the CHS challenge, particularly in Lachnospirales.

### 3.7. Effects of TM and CHS on Cecal Microbiota: Family-Level Composition

Figure 4d illustrates the top 10 bacterial families of the cecal microbiota across all experimental groups. The most prevalent bacterial family were Lachnospiraceae (18.4–35.6%), Ruminococcaceae (17.1–34.2%), Bacteroidaceae (16.5–30.8%), and Lactobacillaceae (5.6–16.6%). Other orders, such as Streptococcaceae, Oscillospiraceae, Rikenellaceae, Erysipelatoclostridiaceae, Butyricicoccaceae, and Clostridia_vadinBB60_group, were present at relatively lower abundances.

Figure 5c shows a significant decrease in the relative abundance of Lachnospiracea in the CHS-TM group (18.4%) compared with other groups (CON: 35.6%; TM: 28.3%; CHS-CON:30. 5).

Figure 5d shows a significant increase in Lactobacillaceaes’ relative abundance in the CHS-challenged groups (CHS-CON: 16.6%; CHS-TM: 10.5%) compared with non-challenged groups (CON: 6%; TM: 5.6%). This result suggests that CHS promotes the growth of beneficial taxes, such as Lactobacillaceae, as a compensatory response to intestinal stress.

### 3.8. Effects of TM and CHS on Cecal Microbiota: Genus-Level Composition

Figure 4e illustrates the top 10 genera of the cecal microbiota across all experimental groups. The most prevalent bacterial genera were non-assigned genera from the *Lachnospiraceae* family (14.8–29%), *Faecalibacterium* (12.3–29.5%), *Bacteroides* (16.5–30.8%), and *Lactobacillus* (4.6–14.2%). Other genera, including *Streptococcus*, *Ruminococcus_torques_group*, *Alistipes*, *Erysipelatoclostridium*, *Blautia*, and *Subdoligranulum*, were detected at comparatively reduced abundance levels.

Figure 5e displays a significant increase in *Lactobacillus’* relative abundance in the CHS-challenged groups (CHS-CON: 14.2%; CHS-TM: 8.6%) compared with non-challenged groups (CON: 4.6%; TM: 4.7%).

## 4. Discussion

This study aimed to investigate the effect of TM (39 °C for 18 h at embryonic days 10–18) on broiler chickens’ body weight, body temperature, and cecal microbiome during CHS on post-hatch days 18 to 22. Previous studies employing a similar TM protocol in Cobb and Hubbard broiler breeds under heat stress conditions reported beneficial effects, including a reduced oxidative stress gene expression, the modulation of inflammatory cytokines and heat shock proteins, and an enhanced antioxidant capacity [41,42].

Our findings indicate that TM-treated chicks exhibited lower body temperatures and altered microbiome compositions compared to control groups. This suggests that TM may induce adaptive changes that enhance heat stress resilience. This reduction in body temperature may be explained by an adaptive mechanism involving the hypothalamus, which is central to regulating body temperature and metabolic processes [23]. During embryogenesis, TM likely alters the development and sensitivity of hypothalamic thermoregulatory centers, resulting in a decreased basal metabolic rate and lower heat production post-hatch [43].

Furthermore, changes at the cellular level, particularly in the mitochondrial function and energy metabolism, could enhance energy efficiency and minimize excessive heat generation, contributing to this observed effect [44]. TM during critical stages of embryonic development can result in prolonged effects on thermotolerance [45]. Previous studies have demonstrated the potential of gut microbiota modulation to enhance thermotolerance, particularly during heat stress [46,47]. A well-balanced gut microbiota is essential for maintaining overall health. It is typically characterized by a higher abundance of beneficial lactic acid bacteria and a lower prevalence of pathogenic species, such as Escherichia coli and other coliforms. This microbial profile supports optimal nutrient absorption and enhances immune function, which may, in turn, contribute to the regulation of body temperature [48,49,50]. However, despite the observed trends, the reduction in the BT in the CHS-TM group was not statistically significant, indicating the need for further investigations into the interplay between TM, gut microbiota, and thermoregulatory mechanisms.

HS negatively affects the feed intake, growth rate, feed efficiency, and overall health in broilers [21]. TM has been shown to enhance broiler weights by promoting muscle growth and development through myoblast proliferation [20,51]. This effect may be mediated by TM-induced modifications in growth factors and muscle marker gene expression, improved mitochondrial activity, and metabolic reprogramming that favors muscle growth. Specifically, TM has been associated with the upregulation of the mitochondrial electron transport chain (ETC) genes and transient receptor potential V2 (TRPV2) in skeletal muscle, elevated growth hormone levels, and insulin-like growth factor, along with the reduced expression of muscle degradation markers such as Atrogin-1 [52,53,54,55]. In our study, the CHS-TM group exhibited a significantly higher BW than the CHS-CON group, which is consistent with previous findings and supports the role of TM in preserving growth performance under thermal stress conditions.

The results of this investigation indicate that TM and CHS did not significantly alter the overall microbial richness or diversity, as there were no statistically significant differences in alpha diversity indexes (Shannon and Chao1) between the treatment groups. This finding aligns with previous studies indicating that heat stress does not significantly impact gut microbial diversity [9]. In general, heat stress has a minimum effect on alpha diversity; however, variations in the literature may arise from several factors, such as the severity and duration of the heat stress, with higher temperatures having a greater effect on microbial diversity [13]. Additionally, genetic variation may contribute to these differences. For example, broilers generally exhibit an increased microbial richness and diversity under heat stress, whereas layers show a decrease [56]. Nonetheless, the non-significant decrease in microbial richness and diversity in the CHS-TM group in this study may warrant further investigation.

The beta diversity analysis is often considered more sensitive than alpha diversity for detecting shifts in microbial community structures between groups, as it evaluates the overall compositional dissimilarity rather than the within-sample richness or evenness [57]. Our study’s alpha diversity indices (Chao1 and Shannon) did not differ significantly between groups, suggesting a stable species richness and evenness across treatments. However, the beta diversity analysis revealed significant differences in microbial community structures between the TM and CON groups, indicating that TM altered the microbial composition. This structural shift may be linked to TM’s influence on embryonic digestive tract development—particularly changes in the cecal pH and epithelial environment—that can selectively promote or inhibit colonization by specific taxa [58]. Conversely, there were no significant differences between the CHS-CON and CON groups, indicating that CHS alone did not affect the control group’s microbial community structure.

One possible mechanism underlying microbiota shifts is the effect of TM on the hypothalamic–pituitary–adrenal (HPA) axis. TM alters the HPA axis function during embryogenesis, which improves the chicken’s ability to manage stress by lowering cortisol levels and promoting the release of chemicals that support gut health [17,59]. In contrast, CHS negatively impacts gut health without TM. Elevated cortisol due to CHS induces inflammation, reduces protective mucus secretion, weakens the intestinal barrier, and disrupts bacterial homeostasis [60,61,62,63].

Developmental stages and embryonic environmental factors critically shape gut microbiota diversity. Tong and Cui [64] highlighted this influence in amphibians, while de Jong and Schokker [65] showed that early-life environmental conditions similarly affect gut microbial diversity in broilers. Among these factors, the incubation temperature plays a pivotal role by inducing epigenetic modifications (e.g., DNA methylation and histone changes) that influence microbial colonization and gut health [66,67]. Specifically, David et al. [66] found that TM during embryogenesis alters histone marks (H3K4me3 and H3K27me3) in chickens’ hypothalamus and muscle, which may contribute to an environmental memory, affecting neurodevelopmental and immune functions. These epigenetic changes are likely to extend to the gut, modulating the microbiota structure.

The peri-hatching period is also essential for the initial microbiota establishment, with long-term health implications [67]. TM may support the colonization by beneficial microbes via these epigenetic mechanisms. Supporting this, an in ovo microbiome modulation using probiotics, prebiotics, or synbiotics has been shown to trigger significant epigenetic alterations in tissues like the liver and spleen, reinforcing the connection between early environmental interventions and microbiota regulation [68,69].

Maintaining a balance of intestinal bacteria is crucial for broilers’ overall health and productivity [70]. The cecum, essential for food digestion and energy extraction and ultimately affects poultry’s intestinal health and development efficiency, is host to the most diverse microbial populations [71,72]. In our study, the chicks’ cecal microbiota was dominated by the phyla Firmicutes and Bacteroidota, which aligns with previous research [11,73].

This reduction in the Firmicutes-to-Bacteroidota ratio in the CHS-TM group suggests that the TM may have influenced the microbial composition in response to CHS. Such shifts in phylum-level abundance could affect the gut microbiota functionality and host metabolism. The CHS-TM group exhibited a higher relative abundance of Bacteroidota. This phylum primarily colonizes the distal intestine, where it plays a crucial role in fermenting otherwise indigestible polysaccharides, thereby contributing to the host’s energy metabolism. The fermentation process produces short-chain fatty acids (SCFAs), a significant energy source, mainly when consumed with a fiber-rich diet [74,75]. Bacteroidota also comprises many typical anaerobic bacteria, which, along with the gut mucosal bioprotective layer, serve as a primary defense mechanism against pathogenic microbes [76]. Conversely, the TM group showed the highest relative abundance of Firmicutes, which is consistent with their involvement with the generation of butyrate and propionate, and SCFAs connected to inflammation management and gut health [77].

Despite these observations, no significant differences in Bacteroidota and Firmicutes abundances were detected between groups at the phylum level. This underscores the need for further investigations to validate these trends and elucidate potential underlying mechanisms. This finding aligns with previous studies indicating that heat stress does not significantly impact the dominant intestinal phyla [63,78].

The relative abundance of the class *Bacilli* was significantly increased in the CHS-challenged groups. This increase was primarily driven by the significant rise in *Lactobacillus* (*c_Bacilli*; *o_Lactobacillales*; *f_Lactobacillaceae* and *g_Lactobacillus*). Previous studies have demonstrated that a *Lactobacillus* supplementation as a probiotic enhances body weight gain, preserves intestinal integrity under stress conditions, and alleviates the adverse effects of heat stress in broilers [79,80,81,82]. Therefore, the increased abundance of *Lactobacillus* observed in this study may represent a compensatory mechanism in response to the intestinal inflammation induced by CHS.

*Lachnospiraceae* were less prevalent in the CHS-TM group than in the other groups. Originating from the ancient Greek words “lachnos” (wooly hair) and “spira” (coil, twist), the *Lachnospiraceae* is a family of *mesophilic*, anaerobic bacteria having a Gram-positive ultrastructure, and live *Lachnospiraceae* supplements have been shown in preclinical in vivo models to enhance gut health and inhibit pathogen colonization [83]. However, despite their role as primary producers of short-chain fatty acids (SCFAs), certain *Lachnospiraceae* taxa have been associated with intestinal and extraintestinal disorders. Notably, some members are related to metabolic syndrome, which could potentially explain the improved growth observed in the CHS-TM group in relation to the decreased abundance of this bacterial taxon [84,85].

This study highlights the potential of TM as a non-invasive and cost-effective strategy for mitigating the adverse effects of CHS on broiler growth and the gut microbiota composition. While TM did not significantly affect alpha diversity, it appeared to influence microbial communities at different taxonomy levels, particularly in response to CHS. In addition, TM may mitigate some of the adverse effects of CHS, as the CHS-TM group exhibited an improved body weight and slightly lower body temperatures compared to the CHS-CON group, suggesting potential protective benefits. Although TM’s impact on thermoregulation was observed, its efficacy in fully counteracting heat stress was limited. Further research using shotgun metagenomics, transcriptomics, or metabolomics is needed to understand microbiome functional mechanisms and long-term effects on poultry health and productivity. Additionally, integrating TM with nutritional interventions, such as prebiotics, probiotics, phytogenic feed additives, and organic acids, presents a promising strategy for mitigating heat stress and optimizing poultry health.

However, our study focused on post-CHS microbiome changes at day 18, leaving two important questions for future research: First, the microbial colonization patterns during the early post-hatch period (days 1–18) preceding the CHS remain uncharacterized, which could reveal how TM shapes the initial gut microbiome establishment. Second, the durability of TM-induced microbiome modifications through peak heat sensitivity periods (day 28) and to market age warrants investigation. Another limitation of this study is the scarcity of the existing literature directly examining the effects of TM on the gut microbiome. While TM has been widely studied in the context of stress physiology, thermoregulation, and growth performance, its impact on microbial community structures remains largely unexplored. This lack of comparative data makes it challenging to contextualize our microbiome findings within a broader framework or to assess whether the observed effects are consistent across different experimental models. Future studies are needed to further investigate the underlying mechanisms linking TM to microbial colonization and functional outcomes.

## 5. Conclusions

This study’s findings indicate that TM applied during embryogenesis results in significant and lasting changes in the gut microbiota composition. TM enhances broilers’ resilience to CHS by modulating the growth of gut bacteria and minimizing body weight loss under heat stress conditions. These findings underscore the practicality of TM as an approach for boosting thermotolerance and improving gut health in poultry production systems.

## Figures and Tables

**Figure 1 animals-15-01677-f001:**
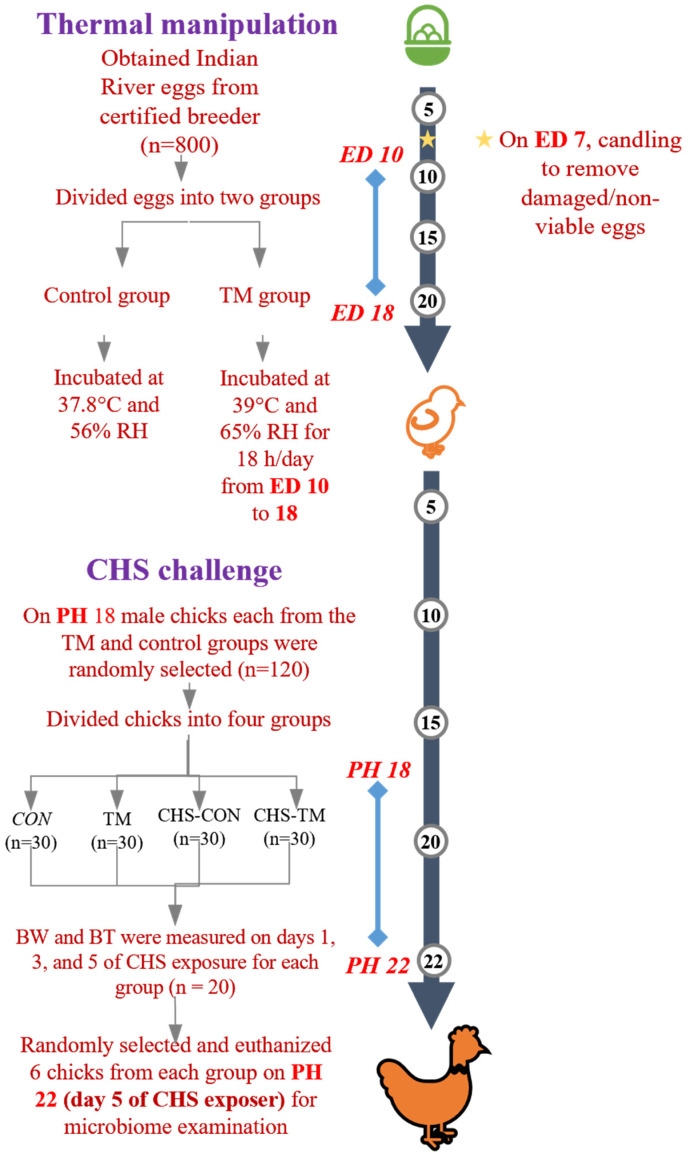
A summary of the experimental design carried out in this study. TM: thermal manipulation, CON: control, RH: relative humidity, CHS: chronic heat stress, ED: embryonic day, and PH: post-hatch.

**Figure 2 animals-15-01677-f002:**
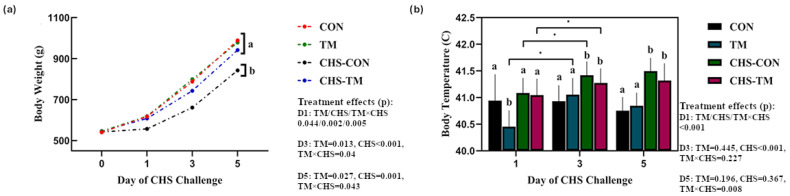
Effects of chronic heat stress (CHS) and thermal manipulation (TM) on broiler chickens (*n* = 6). (**a**) Body weight (BW) gained over five days of the CHS challenge. (**b**) Body temperature (BT) on days 1, 3, and 5 of the CHS challenge. Different superscript letters and * indicate significant differences (*p* < 0.05).

**Figure 3 animals-15-01677-f003:**
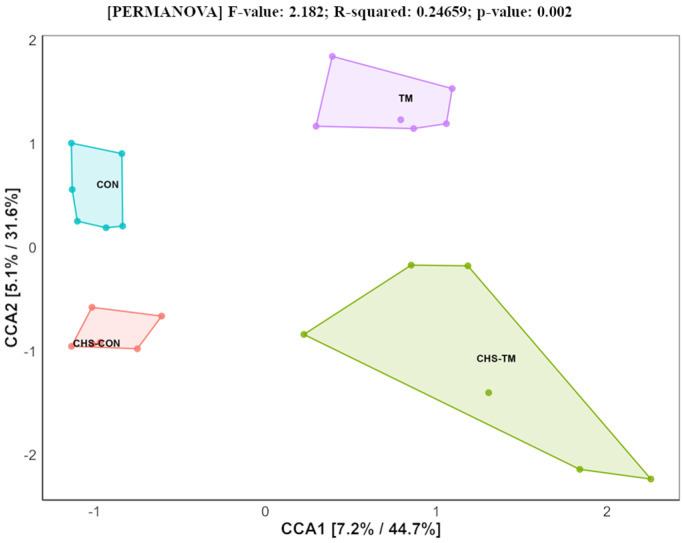
Canonical Correspondence Analysis (CCA) of microbial community composition based on Hellinger transformation.

**Figure 4 animals-15-01677-f004:**
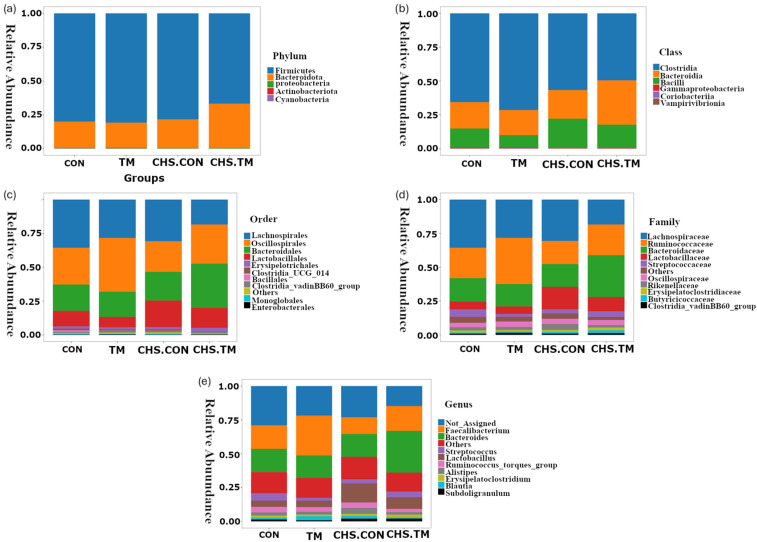
Bar plots showing the relative abundance of the cecum microbiota composition taxa at different taxonomic levels across experimental groups (*n* = 6). (**a**) Phylum, (**b**) class, (**c**) order, (**d**) family, and (**e**) genus. CON = control group; TM = thermal manipulation group; CHS-CON = control with chronic heat stress challenge group; and CHS-TM = thermal manipulation with chronic heat stress challenge group.

**Figure 5 animals-15-01677-f005:**
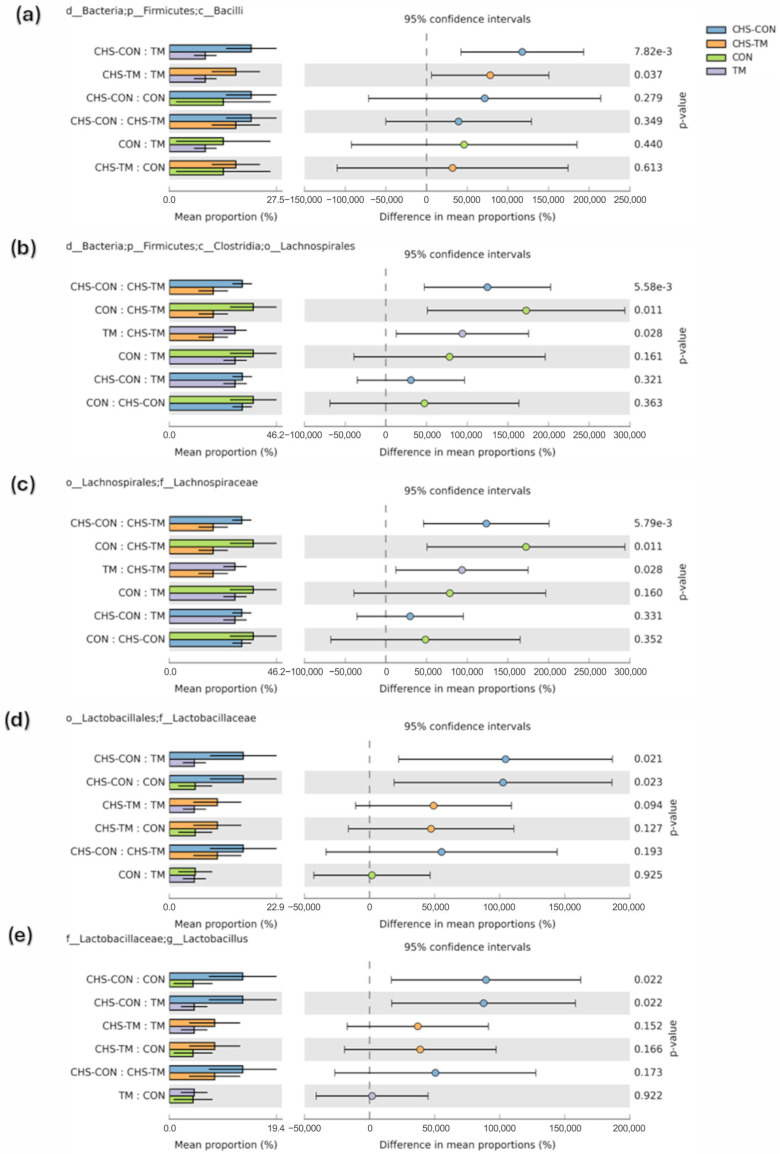
Relative abundance and differences in the mean proportions of bacterial taxa across treatment groups (*n* = 6) with 95% confidence intervals. Taxonomic comparisons include (**a**) Bacilli, (**b**) Lachnospirales, (**c**) Lachnospiraceae, (**d**) Lactobacillaceae, and (**e**) *Lactobacillus*. Significant *p*-values indicate statistically meaningful differences among groups: CON = control group; TM = thermal manipulation group; CHS-CON = control with chronic heat stress challenge group; and CHS-TM = thermal manipulation with chronic heat stress challenge group.

**Table 1 animals-15-01677-t001:** Experimental diet composition based on NRC (1994) guidelines.

Ingredient (% of Diet)	Starter	Grower
Corn	56.80	64.76
Soybean Meal (CP 44%)	35.40	27.85
Fish Meal	1.00	1.00
Soybean Oil	2.31	1.39
Oyster Shell	1.34	1.84
Dicalcium Phosphate	1.53	1.66
Common Salt	0.396	0.326
Vitamin Premix ^a^	0.50	0.50
Mineral Premix ^b^	0.50	0.50
DL-Methionine	0.151	0.055
L-Lysine HCl	0.073	0.119
Total	100	100
Nutrient Composition		
Nutrient	Starter	Grower
AMEn (Kcal/Kg)	2950	2960
Crude Protein (%)	21.203	18.499
Arg (%)	1.365	1.157
Lys (%)	1.208	1.052
Met (%)	0.490	0.360
Met + Cys (%)	0.832	0.666
Ca (%)	0.997	1.200
Available P (%)	0.453	0.467
Na (%)	0.180	0.150

^a^ Vitamin mix (per kilogram of food): Calcium Pantothenate, 19.6 mg; Niacin, 59.4 mg; Pyridoxine, 5.88 mg; Folic Acid, 2 mg; Vitamin B_12_, 0.03 mg; Biotin, 0.2 mg; Choline Chloride, 500 mg; Antioxidant, 2 mg; Vitamin A, 18,000 IU; Vitamin D_3_, 4000 IU; Vitamin E, 36 mg; Vitamin K_3_, 4 mg; Thiamine, 3.5 mg; Riboflavin, 13.2 mg. ^b^ Mineral Premix (per kilogram of food): 198.4 mg of Mn, 169.4 mg of Zn, 100 mg of Fe, 20 mg of Cu, 1.98 mg of I, and 0.4 mg of Se.

**Table 2 animals-15-01677-t002:** Alpha diversity indexes of microbial communities.

Index	CON	TM	CHS-CON	CHS-TM	TM Effect (*p*)	CHS Effect (*p*)	TM × CHS (*p*)
Chao1	149.67 ± 31.7	166.89 ± 24.12	155.67 ± 39.39	138.5 ± 34.92	0.997	0.419	0.219
Shannon	3.73 ± 0.23	3.74 ± 0.31	3.8 ± 0.50	3.4 ± 0.31	0.214	0.333	0.125

Abbreviations: CON = control; TM = thermal manipulation; CHS-CON = control with chronic heat stress challenge; and CHS-TM = thermal manipulation with chronic heat stress challenge. Values represent mean ± SD (*n* = 6). Two-way ANOVA revealed no significant main effects or interactions (all *p* > 0.05).

**Table 3 animals-15-01677-t003:** A PERMANOVA results summary for the difference in the composition of the microbial community.

Pair	F-Value	*p*-Value
TM vs. CON	2.033	0.041
TM vs. CHS-CON	3.4913	0.012
TM vs. CHS-TM	1.5514	0.08
CON vs. CHS-CON	1.0491	0.398
CON vs. CHS-TM	1.3863	0.143
CHS-CON vs. CHS-TM	1.6511	0.087

## Data Availability

All raw sequence data (FASTQ files) and QIIME 2 artifacts, including quality control files, feature tables, taxonomy tables, and rooted phylogenetic trees, related to this study are publicly available and can be obtained at the following link: https://justedujo-my.sharepoint.com/:f:/g/personal/mbvlab_just_edu_jo/EiIiLqseoBhIn6rCb_RnCrQBj3T59obQKOY5x5mWuC6UXA?e=8YLvGV (accessed on 26 May 2025). Furthermore, the NCBI SRA database offers the raw sequencing data under the BioProject ID PRJNA1268758.

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
