# Peer review of "Embryonic Thermal Manipulation Affects Body Performance Parameters and Cecum Microbiome in Broiler Chickens in Response to Post-Hatch Chronic Heat Stress Challenge"

_animals, 2025, doi:10.3390/ani15121677_

Round 1

Reviewer 1 Report

Comments and Suggestions for Authors

The current study has merits since it is one of the first to assess the links between TM and gut microbiota in chickens. while scientifically sound, some issues need to be addressed to improve the quality of the manuscript and increase its relevance to the targeted audience. Indeed authors appears to be expert in thermal manipulation and HS in broilers but may have a limited understanding of microbiota dynamics as it could be noticed in the result and discussion section. specific comments have been attached below.

Abstract

L38: Please specify the duration (hours per day) of the 39°C exposure.

L39: Please provide the standard deviation (SD) for the achieved Chronic Heat Stress (CHS) temperature.

L40: Please provide the mean and standard deviation (SD) for the achieved control temperature.

L42-43: The statement "while CHS-TM exhibited a slight but non-significant BT reduction under heat stress" requires clarification. Compared to which group was this reduction observed? Please specify the comparison group (e.g., CHS without TM?).

L42-43: Additionally, stating "non-significant" is sufficient; describing the reduction as "slight" is unnecessary when statistical significance is reported.

L43: Before presenting the microbiota results, please include a brief introductory phrase to transition the reader.

L44: Please ensure consistent phylum nomenclature throughout the manuscript (e.g., use either Firmicutes/Bacteroidetes or the updated Bacillota/Bacteroidota consistently). Also, ensure that only genus and species names are italicized, not phylum names.

Introduction

L58-59: Please review the entire manuscript for consistent and correct capitalization (Gut is written everywhere).

L59-64: This sentence is overly long and complex. Please break it into shorter sentences and avoid using parentheses for essential information.

Materials and Methods

L116-119: This content appears speculative or interpretive and is better suited for the Discussion section, if necessary, rather than the Materials and Methods.

L150: Broilers typically show increased sensitivity to heat stress (HS) from day 21 onwards, with sensitivity potentially peaking later (e.g., day 28). Please justify the choice of using 18-day-old birds for the HS challenge.

L156: Given the relatively small number of birds per group, please explain the rationale for sampling body weight (BW) rather than assessing BW for all birds within each group.

L162-163: The description implies cecum removal before euthanasia. This procedure must be corrected to state clearly that birds were euthanized before sample collection to comply with ethical animal welfare standards.

L167: The text seems to indicate that both feces and cecal contents were used for microbiota analysis. Please clarify if this is correct. If so, clearly separate the collection and processing methods for each sample type in this section.

L190: Please correct the phrasing. PERMANOVA is a statistical test that is "performed" or "conducted," not "calculated."

L207: Alpha diversity indices often do not meet the assumptions of normality required for a two-way ANOVA (e.g., Gaussian distribution). Consider using a non-parametric alternative suitable for factorial designs, such as the Scheirer-Ray-Hare test. Appropriate non-parametric post-hoc tests (e.g., Dunn's test) should follow. (The 'rcompanion' package in R offers relevant functions).

Results

Figure 3: Please display the p-values for the main effects (e.g., CHS, TM) and their interaction directly on the graph, or clearly in the figure legend.

L216: Were there statistically significant interaction effects between the main factors? Please report the specific p-values for main factors and interactions when discussing the results. This applies throughout the Results section where relevant.

L221: Please report results based on statistical significance. Avoid subjective terms like "slightly lower" or "higher" when describing non-significant differences. State clearly whether differences were significant (and provide p-value) or not significant. Apply this principle throughout the Results section.

Table 2: Please include the p-values for the main effects (TM, HS) and the TM*HS interaction directly within the table.

Figure 4: Consider whether Figure 4 is essential for conveying the results, as the information might be adequately presented elsewhere (e.g., supplementary files).

Figure 5: Please justify the choice of Canonical Correspondence Analysis (CCA) over other potential ordination or visualization techniques (e.g., PCoA, NMDS) in the Methods section. Additionally, consider performing a PERMDISP test to assess homogeneity of multivariate dispersions (the vegan package of R has the betadisper() function for this purpose). The overall p-value from the PERMANOVA test should also be indicated on the plot or in the legend.

L260: The text claims CHS had minimal effects on microbial community structure, but the main effect of CHS (compared to Control) does not appear to be explicitly presented or statistically tested (e.g., via PERMANOVA). Please clarify how this conclusion was reached or present the relevant analysis.

L263: Please use the term "microbial community structure" rather than "composition" when referring to beta diversity analysis results (e.g., from PERMANOVA/CCA).

L269-272: Please specify the statistical test used to compare the relative abundances of bacterial taxa (e.g., presented in Figure 7 or related text) in the Materials and Methods section.

Figure 6: The text within the plot legends is difficult to read. Please increase the font size for better visibility.

L289: This sentence appears interpretive and should be moved to the Discussion section.

Discussion

L350-353: This sentence/passage is too long. Please break it down into shorter, clearer sentences.

L360-362: Please provide citations to support these statements about gene expression or functions, as these were not directly assessed in the current study.

L365-367: This statement is vague. Please elaborate on the specific links observed or hypothesized between the microbiota characteristics under TN conditions and body weight (BW) performance, based on your results or existing literature.

L373: Please provide a citation to support this statement.

L369-383: This section requires significant revision. Instead of defining alpha diversity indices, please focus on interpreting your findings. Specifically, discuss potential reasons why alpha diversity (richness and evenness) did not significantly differ between groups in your study, considering the experimental conditions and relevant literature.

L385-392: Similar to the previous point, this section focuses too much on general concepts of beta diversity analysis. Please revise to specifically discuss and interpret your beta diversity results (e.g., the findings from PERMANOVA, PERMDISP if performed).

L400: The assertion that diversity decreases under heat stress requires substantiation with references. Note that some studies report contrasting findings (e.g., increased or unchanged diversity). Please discuss your results in the context of this existing literature.

L405-407: Please explain the reasoning behind the statement that beta diversity analysis is "more sensitive" to microbial changes compared to alpha diversity in the context of your study or relevant literature.

L415: A key question arising from your results seems to be why TM induced significant microbial changes while chronic HS alone apparently did not (based on L260). Both are "HS exposure" and the former happens during incubation, why such different effects? This requires further discussion and potential explanations.

L428: The discussion mentions the Firmicutes to Bacteroidetes ratio. Please clarify where this ratio was calculated and statistically analyzed, as this analysis does not appear to be presented in the Methods or Results sections. Conclusions cannot be drawn from visual assessment of relative abundance bars alone.

L440-442: The statement linking a higher Firmicutes/Bacteroidetes ratio to better outcomes for the birds needs careful revision and context. This ratio is often associated with obesity or increased energy harvest in mammals. Please clarify why this would be considered beneficial in broiler chickens under these conditions and support your claims with appropriate, context-specific references. Consider reviewing literature specifically on this ratio in poultry and heat stress.

Author Response

Comments 1: L38: Please specify the duration (hours per day) of the 39°C exposure.

Response 1: Thank you for your recommendation. We have revised the abstract section to specify the daily duration of thermal exposure. The sentence is now read in the revised manuscript (Page 1, Line 38):

“Fertile Indian River eggs (n=800) were incubated under control (37.8°C, 56% RH) or TM conditions (39°C, 65% RH for 18 hours per day from embryonic day 10 to 18).”

Comments 2: L39: Please provide the standard deviation (SD) for the achieved Chronic Heat Stress (CHS) temperature.

Response 2: Thank you for this valuable suggestion. The revised text (Page 1, Line 40) now states:

“On post-hatch day 18, male chicks were assigned to either CHS (35±0.5°C for five days) or thermoneutral conditions (24±0.5°C).”

Comments 3: L40: Please provide the mean and standard deviation (SD) for the achieved control temperature.

Response 3: Thank you for this valuable suggestion. The revised text (Page 1, Line 40) now states:

“On post-hatch day 18, male chicks were assigned to either CHS (35±0.5°C for five days) or thermoneutral conditions (24±0.5°C).

Comments 4: L42-43: The statement "while CHS-TM exhibited a slight but non-significant BT reduction under heat stress" requires clarification. Compared to which group was this reduction observed? Please specify the comparison group (e.g., CHS without TM?).

Response 4: Thank you for your helpful observation. We have clarified the comparison group in the revised abstract. The sentence now reads:

“Under thermoneutral conditions, TM chicks had lower BT on day 1 (p<0.05), while CHS-TM exhibited a non-significant BT reduction compared to CHS-CON under heat stress (p>0.05).”

Comments 5: L42-43: Additionally, stating "non-significant" is sufficient; describing the reduction as "slight" is unnecessary when statistical significance is reported.

Response 5: We appreciate the reviewer’s insightful suggestion. We have revised the sentence in the abstract to clarify the comparison group and to remove the redundant descriptor “slight.” The sentence now reads (Page 1, lines 41-42):

“Under thermoneutral conditions, TM chicks had lower BT on day 1 (p<0.05), while CHS-TM exhibited a non-significant BT reduction compared to CHS-CON under heat stress (p>0.05).”

Comments 6: L43: Before presenting the microbiota results, please include a brief introductory phrase to transition the reader.

Response 6: We thank the reviewer for this helpful suggestion. To improve the flow of the abstract, we have added the transitional phrase (Page 1, lines 43-44) “Analysis of the gut microbiome showed that beta diversity analysis (PERMANOVA, p<0.05) indicated distinct microbial shifts.”

Comments 7: L44: Please ensure consistent phylum nomenclature throughout the manuscript (e.g., use either Firmicutes/Bacteroidetes or the updated Bacillota/Bacteroidota consistently). Also, ensure that only genus and species names are italicized, not phylum names.

Response 7: We appreciate the reviewer’s comment. The phylum-level nomenclature in our manuscript reflects classifications derived from the SILVA 138.1 database using the silva-138-99-nb-classifier, which adopts updated taxonomic labels such as Bacteroidota and Firmicutes. We have ensured consistency in phylum-level nomenclature across the manuscript based on this taxonomy. Additionally, we have verified that only genus and species names are italicized, and higher taxonomic levels (e.g., phylum, class, order) are presented in standard font as per scientific conventions.

Comments 8: L58-59: Please review the entire manuscript for consistent and correct capitalization (Gut is written everywhere).

Response 8: Thank you for highlighting this point. We have thoroughly reviewed the manuscript and corrected all instances of inappropriate capitalization. The term “gut” has now been consistently written in lowercase.

Comments 9: L59-64: This sentence is overly long and complex. Please break it into shorter sentences and avoid using parentheses for essential information.

Response 9: We appreciate the suggestion. We have revised the text (Page 2, lines 58-65) as follows:

“Among the various physiological systems affected by HS, the gut (gastrointestinal tract) is especially vulnerable [4]. Both acute and chronic ambient temperature fluctuations can significantly impact gut health and function [5]. HS disrupts normal blood flow by promoting peripheral vasodilation and inducing vasoconstriction in the gut, which leads to hypoperfusion. This impaired circulation damages the intestinal mucosa, increasing permeability and allowing bacteria and toxins to enter the bloodstream. Additionally, HS disrupts the gut microbiome, causing dysbiosis. This imbalance compromises overall health and performance and heightens the risk of pathogenic colonization [5-9].”

Comments 10: L116-119: This content appears speculative or interpretive and is better suited for the Discussion section, if necessary, rather than the Materials and Methods.

Response 10: We appreciate the suggestion. The interpretive content describing prior findings using the TM protocol has been moved from the Materials and Methods section to the Discussion (Page 12, lines 346-349).

Comments 11: L150: Broilers typically show increased sensitivity to heat stress (HS) from day 21 onwards, with sensitivity potentially peaking later (e.g., day 28). Please justify the choice of using 18-day-old birds for the HS challenge.

Response 11: We appreciate the reviewer’s valuable suggestion regarding the long-term effects of TM. We have incorporated the following clarifications into the manuscript:

In Materials and Methods (Page 5, lines 152-156):

“While broiler heat sensitivity typically peaks around day 28, we initiated CHS at day 18 to evaluate TM-induced adaptations before peak sensitivity while avoiding the severe welfare challenges associated with prolonged heat stress in older birds. All environmental conditions and sampling procedures were rigorously standardized across treatment groups.”

In Limitations Section (Page 15, lines 492-497):

“Our study focused on post-CHS microbiome changes at day 18, leaving two important questions for future research: First, the microbial colonization patterns during the early post-hatch period (days 1-18) preceding CHS remain uncharacterized, which could reveal how TM shapes initial gut microbiome establishment. Second, the durability of TM-induced microbiome modifications through peak heat sensitivity periods (day 28) and to market age warrants investigation.

Comments 12: L156: Given the relatively small number of birds per group, please explain the rationale for sampling body weight (BW) rather than assessing BW for all birds within each group.

Response 12: We thank the reviewer for this important observation. BW was assessed in a representative sample of birds rather than the entire group to minimize handling stress and logistical burden. Additionally, this approach helped to avoid pseudoreplication by maintaining the independence of biological replicates.

Comments 13: L162-163: The description implies cecum removal before euthanasia. This procedure must be corrected to state clearly that birds were euthanized before sample collection to comply with ethical animal welfare standards.

Response 13: Thank you for your observation. We have revised the sentence to clarify the correct sequence of procedures by ethical animal welfare standards. The updated text (Page 6, lines 164-165) now reads:

“The chickens were humanely euthanized, and the ceca were aseptically removed using sterile tools.”

Comments 14: L167: The text seems to indicate that both feces and cecal contents were used for microbiota analysis. Please clarify if this is correct. If so, clearly separate the collection and processing methods for each sample type in this section.

Response 14: We appreciate the reviewer’s careful attention. We confirm that only cecal content—not feces—was used for microbiota analysis. To eliminate ambiguity, we revised the text (Page 6, lines 169-170) to read:

“Bacterial DNA was extracted from approximately 250 mg of cecal content using the DNeasy PowerSoil Pro Kit (QIAGEN, CA, USA)”

Comments 15: L190: Please correct the phrasing. PERMANOVA is a statistical test that is "performed" or "conducted," not "calculated."

Response 15: Thank you for pointing this out. We have revised the sentence to accurately reflect the correct terminology. The updated text (Page 6, line 192) now reads:

“MicrobiomeAnalyst performed PERMANOVA analysis based on the Bray-Curtis dis-similarity measures [39].”

Comments 16: L207: Alpha diversity indices often do not meet the assumptions of normality required for a two-way ANOVA (e.g., Gaussian distribution). Consider using a non-parametric alternative suitable for factorial designs, such as the Scheirer-Ray-Hare test. Appropriate non-parametric post-hoc tests (e.g., Dunn's test) should follow. (The 'rcompanion' package in R offers relevant functions).

Response 16: We appreciate the reviewer’s thoughtful suggestion. Before applying the two-way ANOVA, we conducted Shapiro–Wilk tests for normality and Levene’s tests for homogeneity of variances for each alpha diversity metric. The results confirmed that the data met the necessary assumptions for parametric testing. Therefore, we proceeded with the two-way ANOVA analysis. However, we acknowledge the value of non-parametric alternatives in future studies where these assumptions may not be met.

Comments 17: Figure 3: Please display the p-values for the main effects (e.g., CHS, TM) and their interaction directly on the graph, or clearly in the figure legend.

Response 17: We appreciate the reviewer’s suggestion. As requested, we have now displayed the p-values for the main effects (CHS, TM) and their interaction (CHS × TM) directly on the updated version of Figure 3 to enhance clarity and interpretability.

Comments 18: L216: Were there statistically significant interaction effects between the main factors? Please report the specific p-values for main factors and interactions when discussing the results. This applies throughout the Results section where relevant.

Response 18: We appreciate the reviewer’s comment. We have revised the Results section to include the p-values for the main effects (TM and CHS) and their interaction (TM × CHS) derived from the two-way ANOVA.

Comments 19: L221: Please report results based on statistical significance. Avoid subjective terms like "slightly lower" or "higher" when describing non-significant differences. State clearly whether differences were significant (and provide p-value) or not significant. Apply this principle throughout the Results section.

Response 19: Thank you for your valuable suggestion. We have revised the Results section to report only statistically significant findings, avoiding subjective descriptors such as "slightly higher" or "lower" for non-significant differences.

Comments 20: Table 2: Please include the p-values for the main effects (TM, HS) and the TM*HS interaction directly within the table.

Response 20: Thank you for your valuable suggestion. We have revised Table 2 to include the p-values for the main effects (TM, CHS) and their interaction (TM × CHS) directly within the table.

Comments 21: Figure 4: Consider whether Figure 4 is essential for conveying the results, as the information might be adequately presented elsewhere (e.g., supplementary files).

Response 21: Thank you for your suggestion. We agree that Figure 4, which displays rarefaction curves, could be better suited to the supplementary materials. Accordingly, we have moved it to Supplementary Figure S2 and revised the main text to reflect this change while preserving the essential information.

Comments 22: Figure 5: Please justify the choice of Canonical Correspondence Analysis (CCA) over other potential ordination or visualization techniques (e.g., PCoA, NMDS) in the Methods section. Additionally, consider performing a PERMDISP test to assess homogeneity of multivariate dispersions (the vegan package of R has the betadisper() function for this purpose). The overall p-value from the PERMANOVA test should also be indicated on the plot or in the legend.

Response 22: We appreciate the reviewer’s insightful comment. Canonical Correspondence Analysis (CCA), introduced by ter Braak, revolutionized ordination methods by combining correspondence analysis (CA) with regression methodologies, enabling hypothesis testing in addition to exploratory analysis (ter Braak, 1995, Gauch Jr et al., 1981, Gauch Jr, 1982). This advancement made CCA a powerful tool for understanding the influence of environmental factors on community composition, particularly in ecology and microbiome research (Wilmes and Bond, 2004, Xia and Sun, 2023).

In response to your second point, we conducted a PERMDISP analysis to assess the homogeneity of group dispersions. The test revealed no significant differences in dispersion (p > 0.05), supporting the validity of the PERMANOVA results. Additionally, we have now included the overall PERMANOVA p-value in the CCA plot.

Comments 23: L260: The text claims CHS had minimal effects on microbial community structure, but the main effect of CHS (compared to Control) does not appear to be explicitly presented or statistically tested (e.g., via PERMANOVA). Please clarify how this conclusion was reached or present the relevant analysis.

Response 23: We appreciate the reviewer’s observation regarding the interpretation of CHS effects on microbial community composition. To address this, we have clarified the presentation of pairwise PERMANOVA comparisons that include CHS effects. Specifically, comparisons between CON vs. CHS-CON (p = 0.398) and TM vs. CHS-TM (p = 0.08) did not reach statistical significance, indicating that the CHS treatment alone did not significantly alter the overall beta diversity within the respective baseline groups.

These results are now more clearly described in the Results section, and we have revised the text (Page 9, lines 262-264) to state: “Pairwise PERMANOVA comparisons indicated no significant differences in microbial community structure between CHS-CON and CON (p = 0.398), or between CHS-TM and TM (p = 0.08), suggesting a relatively limited influence of CHS on beta diversity. In contrast, significant differences were observed between TM and CON (p = 0.041) and between TM and CHS-CON (p = 0.012), highlighting the more pronounced effect of TM on microbial community structure.”

Comments 24: L263: Please use the term "microbial community structure" rather than "composition" when referring to beta diversity analysis results (e.g., from PERMANOVA/CCA).

Response 24: Thank you for the helpful suggestion. We agree that “microbial community structure “is the more accurate term when referring to beta diversity metrics derived from analyses such as PERMANOVA and CCA. Accordingly, we have revised the manuscript to replace “composition” with “structure.”

Comments 25: L269-272: Please specify the statistical test used to compare the relative abundances of bacterial taxa (e.g., presented in Figure 7 or related text) in the Materials and Methods section.

Response 25: Thank you for your comment. The statistical test used for comparing the relative abundances of bacterial taxa was indeed specified in the Materials and Methods section. Specifically, we stated:

“Using Software for Analyzing Taxonomic or Metabolic Profiles (STAMP), the Kruskal-Wallis H-test was conducted for the relative abundance comparison across subgroups. Standard deviations were visualized using error bars (40).”

To further improve clarity and ensure that the test is explicitly linked to the results shown in Figure 6, we have revised the sentence slightly (Page 6, lines 195-197) to read:

“For comparing relative abundances of bacterial taxa (e.g., Figure 6), the Kruskal-Wallis H-test was performed using STAMP software, and standard deviations were visualized with error bars [40].”

Comments 26: Figure 6: The text within the plot legends is difficult to read. Please increase the font size for better visibility.

Response 26: Thank you for your helpful observation. In response to your comment, we have increased the font size of the legends in Figure 6 (now Figure 5) to enhance readability and ensure better visual clarity.

Comments 27: Figure 6: The text within the plot legends is difficult to read. Please increase the font size for better visibility.

Response 27: Thank you for your observation. We agree with your suggestion. The sentence at Line 289 has been removed from the Results section, as it is already appropriately addressed in the Discussion section.

Comments 28: L289: This sentence appears interpretive and should be moved to the Discussion section.

Response 28: Thank you for your observation. We agree with your suggestion. The sentence at Line 289 has been removed from the Results section, as it is already appropriately addressed in the Discussion section.

Comments 29: L350-353: This sentence/passage is too long. Please break it down into shorter, clearer sentences.

Response 29: Thank you for the helpful suggestion. We have revised the sentence to improve clarity and readability by dividing it into shorter, more concise statements. The revised text (Page 13, lines 363-368) now reads:

“A well-balanced gut microbiota is essential for maintaining overall health. It is typically characterized by a higher abundance of beneficial lactic acid bacteria and a lower prevalence of pathogenic species such as Escherichia coli and other coliforms. This microbial profile supports optimal nutrient absorption and enhances immune function, which may, in turn, contribute to the regulation of body temperature [48-50].”

Comments 30: L360-362: Please provide citations to support these statements about gene expression or functions, as these were not directly assessed in the current study.

Response 30: Thank you for this important observation. We agree with the need to clearly attribute findings not directly measured in our study. We have revised the sentence to better reflect that these mechanisms are based on previous studies and have cited the relevant literature accordingly. The revised text (Page 13, lines 373-379) now reads:

“This effect may be mediated by TM-induced modifications in growth factors and muscle marker gene expression, improved mitochondrial activity, and metabolic reprogramming that favors muscle growth. Specifically, TM has been associated with the upregulation of the mitochondrial electron transport chain (ETC) genes and transient receptor potential V2 (TRPV2) in skeletal muscle, elevated growth hormone levels, and insulin-like growth factor, along with reduced expression of muscle degradation markers such as Atrogin-1 [52-55].”

Comments 31: L365-367: This statement is vague. Please elaborate on the specific links observed or hypothesized between the microbiota characteristics under TN conditions and body weight (BW) performance, based on your results or existing literature.

Response 31: We appreciate the reviewer’s comment and agree that the link between microbial characteristics and BW performance requires clarity. Upon review, we found that this concept is more comprehensively addressed in the subsequent paragraph discussing the relative abundance of Lachnospiraceae and its association with growth performance. Therefore, we have removed the earlier sentence to avoid redundancy and to maintain a more coherent and focused discussion. Thank you for helping us improve the clarity and logical flow of our manuscript.

Comments 32: L373: Please provide a citation to support this statement.

Response 32: Thank you for the helpful suggestion. We have revised the statement accordingly (supplementary material) and added a more appropriate citation to support the role of microbial diversity as an indicator of environmental and host-related changes, as well as its central importance in microbiome research.

Comments 33: L369-383: This section requires significant revision. Instead of defining alpha diversity indices, please focus on interpreting your findings. Specifically, discuss potential reasons why alpha diversity (richness and evenness) did not significantly differ between groups in your study, considering the experimental conditions and relevant literature.

L385-392: Similar to the previous point, this section focuses too much on general concepts of beta diversity analysis. Please revise to specifically discuss and interpret your beta diversity results (e.g., the findings from PERMANOVA, PERMDISP if performed).

Response 33: Thank you for the constructive feedback. We have moved the general descriptions of diversity metrics and rarefaction to the Supplementary Material to streamline the main text. In the revised manuscript, we now focus on interpreting the observed alpha and beta diversity results in the context of our experimental conditions and relevant literature

Comments 34: L400: The assertion that diversity decreases under heat stress requires substantiation with references. Note that some studies report contrasting findings (e.g., increased or unchanged diversity). Please discuss your results in the context of this existing literature.

Response 34: We appreciate the reviewer’s feedback. In response, we have removed the sentence stating, “On the other hand, there is a decrease in diversity under heat stress,” to avoid overgeneralization. Additionally, we have acknowledged the limited availability of literature on the microbiome effects of TM as a key limitation of our study (page 15, lines 497-504). This clarification highlights both the novelty of our findings and the need for further research to better understand the relationship between TM and gut microbial dynamics.

Comments 35: L405-407: Please explain the reasoning behind the statement that beta diversity analysis is "more sensitive" to microbial changes compared to alpha diversity in the context of your study or relevant literature.

Response 35: Thank you for your comment. We have revised the paragraph (Page 13, lines 395-399) to explain that beta diversity evaluates between-group differences in microbial composition, which can reveal treatment-induced structural changes not captured by alpha diversity metrics that focus on within-sample diversity. In our study, only beta diversity detected significant differences between TM and CON groups, while alpha diversity remained unchanged. This observation aligns with previous literature and highlights the higher sensitivity of beta diversity analysis in certain experimental contexts [58].

Comments 36: L415: A key question arising from your results seems to be why TM induced significant microbial changes while chronic HS alone apparently did not (based on L260). Both are "HS exposure" and the former happens during incubation, why such different effects? This requires further discussion and potential explanations.

Response 36: We thank the reviewer for this insightful question. In response, we have expanded the Discussion (Page 14, line 414-429) to address why TM during embryogenesis induced significant microbial shifts, while chronic heat stress (CHS) applied post-hatch did not. The revised text provides a mechanistic explanation, highlighting the importance of developmental timing, modulation of the hypothalamic-pituitary-adrenal (HPA) axis, and TM-induced epigenetic programming, which collectively shape gut microbiota during critical windows of microbial colonization. This expanded discussion is now included in the revised manuscript.

Comments 37: L428: The discussion mentions the Firmicutes to Bacteroidetes ratio. Please clarify where this ratio was calculated and statistically analyzed, as this analysis does not appear to be presented in the Methods or Results sections. Conclusions cannot be drawn from visual assessment of relative abundance bars alone.

Response 37: We thank the reviewer for this important observation. In response, we have now explicitly reported the calculation and statistical analysis of the Firmicutes-to-Bacteroidota (F/B) ratio in the Results section (Page 9, lines 278-279). The ratio was calculated at the sample level and analyzed using the Kruskal-Wallis H-test. No significant differences were found among groups (p = 0.12). This clarification has also been reflected in the updated text of the Results section to ensure transparency and methodological rigor.

Comments 38: L428: The discussion mentions the Firmicutes to Bacteroidetes ratio. Please clarify where this ratio was calculated and statistically analyzed, as this analysis does not appear to be presented in the Methods or Results sections. Conclusions cannot be drawn from visual assessment of relative abundance bars alone.

Response 38: Thank you for your comment. We agree that the interpretation of the Firmicutes-to-Bacteroidota (F/B) ratio requires species-specific context and cautious interpretation. To avoid potential overgeneralization and unsupported claims, we have removed the sentence stating that a higher F/B ratio suggests potential benefits for growth performance.

Comments 39: L440-442: The statement linking a higher Firmicutes/Bacteroidetes ratio to better outcomes for the birds needs careful revision and context. This ratio is often associated with obesity or increased energy harvest in mammals. Please clarify why this would be considered beneficial in broiler chickens under these conditions and support your claims with appropriate, context-specific references. Consider reviewing literature specifically on this ratio in poultry and heat stress.

Response 39: Thank you for your comment. We agree that the interpretation of the Firmicutes-to-Bacteroidota (F/B) ratio requires species-specific context and cautious interpretation. To avoid potential overgeneralization and unsupported claims, we have removed the sentence stating that a higher F/B ratio suggests potential benefits for growth performance.

GAUCH JR, H. G. 1982. Noise reduction by eigenvector ordinations. Ecology, 63, 1643-1649.

GAUCH JR, H. G., WHITTAKER, R. H. & SINGER, S. B. 1981. A comparative study of nonmetric ordinations. J. Ecol., 135-152.

TER BRAAK, C. J. F. 1986. Canonical Correspondence Analysis: A New Eigenvector Technique for Multivariate Direct Gradient Analysis. Ecology, 67, 1167-1179.

TER BRAAK, C. J. F. 1995. Ordination.

WILMES, P. & BOND, P. L. 2004. The application of two-dimensional polyacrylamide gel electrophoresis and downstream analyses to a mixed community of prokaryotic microorganisms. Environ. Microbiol., 6, 911-20.

XIA, Y. & SUN, J. 2023. Alpha Diversity. In: XIA, Y. & SUN, J. (eds.) Bioinformatic and Statistical Analysis of Microbiome Data. Cham: Springer International Publishing.

Reviewer 2 Report

Comments and Suggestions for Authors
  1. The authors used "Gut" with capital "G" throughout the manuscript. There is no reason to use the capital, and it should be changed to "gut". Please make the changes throughout the manuscript.
  2.  Line 91-92: to improve the readability, please change the order of the parameters measured according to the order in which they were presented in this manuscript: body weight, body temperature, and cecal microbiome
  3. Line 113: TM was done for 18 hr per day. This line says it was done from 12:00 AM to 8:00 AM. Then TM was done 16 hrs. Please check and correct. 
  4. Line 129-130: Line 136 says the feed was provided ad libitum. But this line says, "before meals were saved". Please make appropriate change.  
  5. Line 131: change table 1 to Table 1. 
  6. Line 150: I think "On" should be removed. 
  7. Figure 2. What is shown here is a quite standardized procedure. I think this should be moved to Supplementary Materials, unless its current location can be justified otherwise. 
  8. Line 222: "at day 19" means the second day of CHS challenge? On the 2 graphs in Figure 3, the time on X axis was shown as Day 0, 1, 3, and 5. It would be helpful to tie "day 19" to the days in the graph. 
  9. Figure 4 is also a part of the standard analysis of 16S rRNA gene data. It should be moved to Supplementary Materials. 
  10. Line 474: change "shut gun" to "shotgun".
  11. I search the literature on the similar subject and found there are quite many publications already on TM and its impact on HS response. I also found the authors already cited many of them. It seems that the authors emphasized the unique contribution of this study is its inclusion of gut microbiome data. However, the main focus of the study is TM and its impact on HS. I think that the authors should make it clear in what way the result of the current study is similar or dissimilar to previous studies on TM and HS. 

Author Response

Comments 1: The authors used "Gut" with capital "G" throughout the manuscript. There is no reason to use the capital, and it should be changed to "gut". Please make the changes throughout the manuscript.

Response 1: We appreciate the reviewer’s meticulous attention to detail. We have revised the manuscript to use "gut" (lowercase) consistently.

Comments 2: Line 91-92: to improve the readability, please change the order of the parameters measured according to the order in which they were presented in this manuscript: body weight, body temperature, and cecal microbiome.

Response 2: We thank the reviewer for this helpful suggestion to improve manuscript consistency. We have revised the text (Page 7, line 92) to present the parameters in the same order they appear in subsequent sections: body weight, body temperature, and cecal microbiome.

Comments 3: Line 113: TM was done for 18 hr per day. This line says it was done from 12:00 AM to 8:00 AM. Then TM was done 16 hrs. Please check and correct.

Response 3: We appreciate the reviewer’s attention to this important clarification. We have revised the description to accurately reflect that thermal manipulation was applied for 18 hours per day, specifically from 12:00 PM (noon) to 8:00 AM the following morning, during embryonic days 10–18.

Comments 4: Line 129-130: Line 136 says the feed was provided ad libitum. But this line says, "before meals were saved". Please make appropriate change.

Response 4: We appreciate the reviewer’s attention to this inconsistency. We have revised the sentence to maintain consistency with the ad libitum feeding protocol. The corrected sentence now reads: “Each group was weighed up in the morning.” (Page 5, line 127)

Comments 5: Line 131: Change table 1 to Table 1.

Response 5: We thank the reviewer for this observation. The text has been revised to correctly capitalize the table reference. The sentence now reads: “…standard nutrient requirements (Table 1)—adapted from Shakouri and Malekzadeh [30].”

Comments 6: Line 150: I think "On" should be removed.

Response 6: We thank the reviewer for this helpful suggestion. The word "On" has been removed for grammatical accuracy. The corrected sentence now reads: “The CHS challenge was performed on the 18th post-hatch day.”

Comments 7: Figure 2. What is shown here is a quite standardized procedure. I think this should be moved to Supplementary Materials, unless its current location can be justified otherwise.

Response 7: Thank you for your valuable feedback. As per your suggestion, we have moved Figure 2 to the Supplementary Materials section.

Comments 8: Line 222: "at day 19" means the second day of CHS challenge? On the 2 graphs in Figure 3, the time on X axis was shown as Day 0, 1, 3, and 5. It would be helpful to tie "day 19" to the days in the graph.

Response 8: Thank you for your insightful comment. We agree that the reference to “day 19” could benefit from clearer alignment with the timeline presented in Figure 3. We have revised the text accordingly to make this temporal relationship explicit, ensuring better clarity for the reader. Revised sentence (Page 7, lines 210-230): “Figure 3a illustrates changes in BW over time following the CHS challenge. On Day 0 of CHS (18 days of age), all groups had comparable initial weights (~540 g). By Day 5 post-CHS, statistically significant effects of TM (p = 0.027), CHS (p = 0.001), and their interaction (p = 0.043) were observed. These findings indicate that both TM and CHS independently influenced the final BW and that their interaction also had a sig-nificant impact. At this time point, the non-challenged groups (CON and TM) reached higher final BW values (988.8 g and 978.1 g, respectively), while the CHS-TM group exhibited a significantly greater BW (941.8 g) than the CHS-CON group (842.3 g), sug-gesting a mitigating effect of TM on CHS-induced growth reduction.

Figure 3b demonstrates body temperature changes among the experimental groups in response to the CHS challenge. On Day 1 post-CHS (18 days of age), a statis-tically significant effect of TM (p < 0.001), CHS (p < 0.001), and their interaction (p < 0.001) was observed, with the TM group showing markedly lower BT values compared to the CON group. During Days 3 and 5 post-CHS (20, 22 days of age), both CHS-exposed groups (CHS-CON and CHS-TM) exhibited significant BT elevations, consistent with the stress response. On Day 3, CHS exerted a significant main effect (p < 0.001), but TM (p = 0.445) and the interaction (p = 0.227) were not statistically sig-nificant. By Day 5, a significant interaction between TM and CHS was detected (p = 0.008), although the main effects of TM (p = 0.196) and CHS (p = 0.367) were not indi-vidually significant. Furthermore, the CHS-TM group maintained non-significantly lower BT values than CHS-CON on Days 3 and 5.”

Comments 9: Figure 4 is also a part of the standard analysis of 16S rRNA gene data. It should be moved to Supplementary Materials.

Response 9: Thank you for your helpful observation. We acknowledge that Figure 4 represents a standard component of 16S rRNA gene data analysis. In response to your suggestion, we have moved Figure 4 to the Supplementary Materials (now Figure S2) to streamline the main manuscript and focus on the novel findings. We appreciate your guidance in improving the clarity and organization of the manuscript.

Comments 10: Line 474: change "shut gun" to "shotgun".

Response 10: Thank you for pointing out the typographical error. We have corrected "shut gun" to "shotgun" as suggested.

Comments 11: I search the literature on the similar subject and found there are quite many publications already on TM and its impact on HS response. I also found the authors already cited many of them. It seems that the authors emphasized the unique contribution of this study is its inclusion of gut microbiome data. However, the main focus of the study is TM and its impact on HS. I think that the authors should make it clear in what way the result of the current study is similar or dissimilar to previous studies on TM and HS.

Response 11: Thank you for this valuable comment. We have revised the Discussion section (Page 13, lines 380-383) to clarify how our findings relate to previous studies on TM and HS. Specifically, we noted that the higher body weight observed in the CHS-TM group is consistent with earlier reports showing that TM promotes muscle development and improves growth performance under HS conditions. These effects have been attributed to TM-induced changes in growth factor signaling, mitochondrial activity, and muscle gene expression. By aligning our results with these established physiological outcomes, we strengthen the context for our findings while emphasizing the novel contribution of microbiome data in this study. Additionally, we now address a key limitation of our study (Page 15, lines 498-505)—the scarcity of microbiome-specific research on TM, which further emphasizes the novelty and relevance of our work.

Round 2

Reviewer 1 Report

Comments and Suggestions for Authors

The authors appropriately revised the manuscript. Congratulations on the work done and the rigor in addressing shortcomings. I only have two more requests:

Figure 5: The font and legend of the figure axis are still too small to read easily. Please increase them to match the size of the text in the manuscript.

The study limitations should be the last paragraph of the discussion, and before the conclusion.

Author Response

Comment 1: Figure 5: The font and legend of the figure axis are still too small to read easily. Please increase them to match the size of the text in the manuscript.

Response 1: Thank you for your feedback. We have revised Figure 5 by increasing the font size of the axis labels, tick marks, and legend to ensure consistency with the manuscript text. The updated version now provides improved readability.

Comment 2: The study limitations should be the last paragraph of the discussion, and before the conclusion.

Response 2: We thank the reviewer for their valuable suggestion. As recommended, we have moved the study limitations to the end of the Discussion section, immediately before the Conclusion.

Reviewer 2 Report

Comments and Suggestions for Authors

The authors responded to and addressed all of my concerns thoroughly. Now I believe this manuscript is ready to be published in the current form. 

Author Response

Comment 1: The authors responded to and addressed all of my concerns thoroughly. Now I believe this manuscript is ready to be published in the current form.

Response 1: Thank you very much for your kind words